# Quality of Life and Physical Performance in Patients with Obesity: A Network Analysis

**DOI:** 10.3390/nu12030602

**Published:** 2020-02-26

**Authors:** Riccardo Dalle Grave, Fabio Soave, Antonella Ruocco, Laura Dametti, Simona Calugi

**Affiliations:** Department of Eating and Weight Disorders, Villa Garda Hospital, 37138 Garda (VR), Italy; info@attiviperstarbene.it (F.S.); anto82ruocco@gmail.com (A.R.); lauradametti96@gmail.com (L.D.); si.calugi@gmail.com (S.C.)

**Keywords:** obesity, physical performance, network analysis, vitality, mental health

## Abstract

Background: The aim of this study was to investigate the interconnections between specific quality-of-life domains in patients with obesity and high or low physical performance using a network approach. Methods: 716 consecutive female and male patients (aged 18–65 years) with obesity seeking weight-loss treatment were included. The 36-item Short Form Health Survey (SF-36) and the six-minute walking test (6MWT) were used to assess quality of life and physical performance, respectively. The sample was split into two groups according to the distance walked in the 6MWT. Network structures of the SF-36 domains in the two groups were assessed and compared, and the relative importance of individual items in the network structures was determined using centrality analyses. Results: 35.3% (*n* = 253) of participants covered more distance than expected, and 64.7% (*n* = 463) did not. Although low-performing patients showed lower quality of life domain scores, the network structures were similar in the two groups, with the SF-36 Vitality representing the central domain in both networks. Mental Health was a node with strong connections in patients who walked less distance. Conclusions: These findings indicate that psychosocial variables represent the most influential and interconnected features as regards quality of life in both groups.

## 1. Introduction

Obesity is a condition characterized by an excessive accumulation of fat in adipose tissue; it is linked to an increased risk of chronic diseases, disability, and mortality [1], and is also often associated with poor physical fitness levels, e.g., muscle strength [2], and cardiorespiratory fitness [3]. Moreover, both obesity and physical performance are associated with quality of life. Indeed, a recent systematic review found that in all populations examined, obesity was associated with a significantly worse generic and obesity-specific quality of life [3]. Furthermore, significant weight loss after a bariatric surgery or non-bariatric interventions has been associated with improvements in quality of life [4]. Some evidence also supports a link between quality of life and physical fitness in adolescent patients with obesity, and a recent study indicated cardiorespiratory fitness as the main mediator in the relationship between body mass index (BMI) and quality of life [5]. However, this relationship requires a more in-depth investigation in adults.

Understanding whether specific aspects of quality of life are more prominent or strongly interlinked in patients with obesity with different levels of physical performance is relevant to the design of targeted interventions to promote optimum weight management, and may require innovative methods of investigation, such as network analysis—a novel way of representing variables as complex dynamic systems of interacting variables. The inspection of networks elucidates the extent to which items belonging to the same construct are connected to each other, and the strength of their reciprocal relationships. Although in the majority of applications network analysis typically used to be limited to determining a network structure in a single population, recently the focus has shifted from single-population studies to the research comparing network structures from different subpopulations [6]. To this end, specific tests have been developed [7] to examine whether the network structure is identical across subpopulations, whether specific correlations differ in strength between subpopulations, and whether the overall connectivity is equal across subgroups.

Network analysis had never before been used to examine the empirical relationships between quality of life domains in patients with obesity, and the aim of the present study was therefore to use a network approach to provide benchmark data on the interconnections between specific health and psychological features of the quality of life in patients with high or low levels of physical performance seeking treatment for obesity.

## 2. Materials and Methods

Participants were recruited from consecutive referrals by family doctors to the rehabilitative treatment programs for obesity at the inpatient unit of the Villa Garda Hospital Department of Eating and Weight Disorders during the years 2016–2019. Patients were eligible for this study if they were aged between 18 and 65 years, had a BMI ≥ 30.0 kg/m^2^, and at least one weight loss-responsive comorbidity (i.e., type 2 diabetes, cardiovascular disease, sleep apnea, severe joint disease, two or more cardiovascular risk factors), as defined by Adult Treatment Panel III [8]. The criteria for exclusion were pregnancy or lactation, medications that affect body weight, medical comorbidities associated with weight loss, severe psychiatric disorders (i.e., bulimia nervosa, acute psychotic disorders, substance use disorders), use of a walker, and the need in assistance/support with walking.

As per the Italian National Health System’s National ethical guidelines, this study was classed as a routine service assessment rather than research per se, as all the procedures used for treatment and assessment were performed as routine clinical practice, and therefore no ethical clearance was necessary. That being said, each patient provided written informed consent to the collection and processing of their anonymous clinical data in the service-level research setting.

All data were collected on the second day of admission to the programs. Specifically, BMI was determined using the standard formula of body weight (kg) divided by height (m^2^) following measurement of body weight and height using medical weighing scales (Seca Digital Wheelchair Scale Model 664, Hamburg, Germany) and a stadiometer (Wall-Mounted Mechanical Height Rod Model 00051A; Wunder, REA (MI), Italy), respectively. The scale was calibrated for accuracy by an external accredited laboratory every two months. For the purposes of these measurements, participants were weighed in the morning (12 h after eating) wearing only lightweight clothes and no shoes and standing with minimal movement with hands by their sides. Body weight was measured once for each participant to the nearest 0.1 kg.

Physical performance was assessed by means of the six-minute walking test (6MWT) [9] according to international guidelines [10]. The 6MWT was performed along a 20 m long corridor in the department, marked with tape on the floor every 2 m; starting and finishing points were also marked on the floor in a similar fashion. Before the start and at the end of each test, pulse, respiratory rate, and oxygen saturation were measured. The patients were instructed to walk as fast as they could, but were allowed to stop or rest during the test if necessary. All participants concluded the test without breaks. The specific reference equation for predicting distance walked in six minutes in adult subjects with obesity [9] was used to assess the difference between the predicted and real 6MWT scores. The patients walking as far as or farther than predicted were included in Group H (i.e., obesity with a higher 6MWT score than expected), and the patients walking less than predicted were allocated to Group L (i.e., obesity with a lower 6MWT score than expected).

The quality of life was assessed using the validated Italian version of the Short Form-36 (SF-36)—a generic health related quality-of-life questionnaire [11,12]. The SF-36 incorporates questions about (role) functioning and satisfaction with various life domains; it consists of 36 questions, and assesses four domains related to the physical component of quality of life (Physical Functioning, Physical Role Functioning, Bodily Pain, General Health Perception), and four domains related to the mental component (Vitality, Social Functioning, Emotional Role Functioning, and Mental Health). SF-36 scale scores range from 0 to 100; a higher score indicates a better quality of life.

### Statistical Analysis

Variables are presented as means and standard deviations, or frequencies and percentages, as appropriate. Either the *t*-test or the chi-squared test was used to compare Group L and Group H, as appropriate. Network analysis was performed on the 8 SF-36 domain scores for each group, thereby creating a graphical representation of the interconnections between SF-36 domains; domains are depicted as nodes, while their intercorrelations are represented as lines, or “edges”—the thicker and more saturated the edge, the stronger the correlation. The network display is based on an algorithm [13] that places strongly associated nodes at the center of the network and weakly associated nodes at the periphery. To reduce the number of false-positive edges, the Least Absolute Shrinkage and Selection Operator (LASSO) was applied. It estimates small or unstable correlations as zero, and thereby creates a conservative model; this way, the network edges that are less likely to be genuine are removed, and the network is easier to interpret.

Once a collection of networks had been obtained, we minimized the Extended Bayesian Information Criterion (EBIC) [14] to optimize their fit; this process is a particularly effective means of revealing the true network structure [15,16], especially when the generating network is sparse (i.e., does not contain many edges).

To quantify the importance of each node in the network, we then calculated the betweenness, closeness, and strength centrality indices. The betweenness denotes the number of times a specific node acts as a bridge along the shortest path between two nodes, while the closeness measures the number of direct and indirect links between each node and the others; the strength of these inter-node connections is expressed as the degree. [17]. Each of these indices were normalized (mean = 0, and standard deviation (SD) = 1), so that an index value of > 1 indicates that it is > 1 SD from the mean.

Data management and descriptive analyses were performed using SPSS version 26, and the network analysis—using the JASP version 0.10.2 statistical software (Department of Psychological Methods University of Amsterdam, Amsterdam, The Netherlands, https://jasp-stats.org/).The R-package NetworkComparisonTest was used to test the invariant network structure, the invariant edge strength, and the invariant global strength between subgroups [7].

## 3. Results

### 3.1. Patient characteristics

Of the 716 patients recruited, 35.3% (*n* = 253) covered more distance in the 6MWT than predicted, and 64.7% (*n* = 463) did not. On the basis of these distances, the patients were allocated to Groups H and L, respectively. The two groups had similar age, BMI and waist circumference. However, Group H patients had greater body weight and higher scores in all SF-36 domains than those in Group L. A greater proportion of males than females reached a higher 6MWT score than expected (Table 1).

### 3.2. Network structure in Group L and Group H

The network analysis was carried out on the overall sample, which included 253 Group H patients and 463 Group L patients. The network structure confirmed that SF-36 physical and mental components, colored in black and white, respectively, comprised two distinct clusters in both Groups (Figure 1). Groups H and L displayed similar values for the maximum difference in all of the edge weights of the networks (M = 0.30, *p* = 0.11). Moreover, the difference in global strength between the networks was not significant (S = 0.18, *p* = 0.82).

Concerning the centrality of SF-36 domains, two domains played a key role. In Group H, Physical Functioning and Vitality had the highest betweenness (directly connecting more items with each other) and closeness (direct and indirect connections with other items), and Vitality had the highest degree (stronger links with other items). On the other hand, in Group L, Emotional Role Functioning and Physical Role Functioning had the highest betweenness, whereas Vitality had highest closeness, and Vitality and Mental Health the highest degrees (stronger links with other items).

## 4. Discussion

This study aimed to evaluate the interconnections between quality-of-life domains in patients with obesity and either low or high physical performance levels using a network approach. This innovative analysis revealed three main findings. Firstly, about two-thirds of patients with obesity walked a smaller distance than expected. This could be attributed to the severity of clinical features in our sample, which was comprised of patients seeking treatment for obesity in an inpatient setting, and could indicate that their reduced functional capacity was due to comorbid conditions associated with obesity [9].

The second finding concerns the differences between the two groups. As expected, the lower-performing patients had a lower quality of life than those who walked farther than predicted, confirming that physical functioning and quality of life are associated in both the physical and mental domains of the latter.

Our third finding indicated that the network structures of low- and high-performing patients seeking treatment for obesity are invariant. This indicates that the key elements for evaluating the quality of life in a person with obesity are similar, regardless of their physical performance level. In both networks, Vitality (a domain including items investigating pep/life, energy, worn out, tired) plays a key role and represents the domain with the strongest connections with all the other domains, indicating the importance of this variable in the perception of quality of life. In low-performing patients, Mental Health (a domain including items investigating nervous, down in dumps, peaceful, blue/sad, happy) was found to be a key variable, too, suggesting that patients with low physical performance tend to judge their quality of life based mainly on psychological variables, and seem less interested in physical variables. This could, in part, explain the less attention to maintaining good physical performance in this subgroup of patients with obesity.

The study has two main strengths. Firstly, to our knowledge, it is the first to apply network analysis to investigate the relationships between quality of life domains in patients with obesity, and to explore the network structure and strength of relationships between quality of life domains as related to lower and higher physical performance levels. Secondly, the fact that we used the 6MWT to measure performance means that the study would be easy to replicate. Testing the ability to walk a distance is a quick and inexpensive measure of physical function, and an important component of quality of life, since it reflects the capacity to undertake day-to-day activities.

However, the study also has certain weaknesses. Firstly, it was a cross-sectional study measuring quality of life during a single examination session, and we cannot therefore draw conclusions about the association between physical performance and quality of life in the management of obesity over time. Secondly, while we have routinely measured pulse, oxygen, and respiratory rates during the 6MWT, we have not collected these data in the data set, and therefore we do not have accurate information about these variables of physical fitness. Thirdly, generalizing these study’s findings beyond this inpatient population should be attempted with caution, because our sample may not be representative of patients with obesity seeking treatment in other settings, such as outpatient treatment, or subjects with obesity not seeking treatment.

## 5. Conclusions

Network comparisons provided interesting insight into the most interlinked quality of life domains in patients with obesity and low and high physical performance levels, revealing similar network structures, with Vitality playing a central role among quality of life variables. Moreover, in patients with obesity and low physical performance levels, Mental Health is a central variable, indicating that psychological aspects should be considered in defining quality of life in patients with low physical performance levels. Knowledge of these aspects can provide a useful guide for clinicians, suggesting the use of psychosocial interventions and improving the importance of physical fitness aspects in obesity management, especially in patients with low physical performance. Future studies should contribute to clarifying the relationship between quality of life and physical performance using new statistical approaches, including network analysis. Moreover, well-conducted longitudinal clinical trials and intervention studies should be performed to evaluate the effect of associating strategies to improve mental health on the standard weight management in improving physical fitness and quality of life of patients with obesity.

## Figures and Tables

**Figure 1 nutrients-12-00602-f001:**
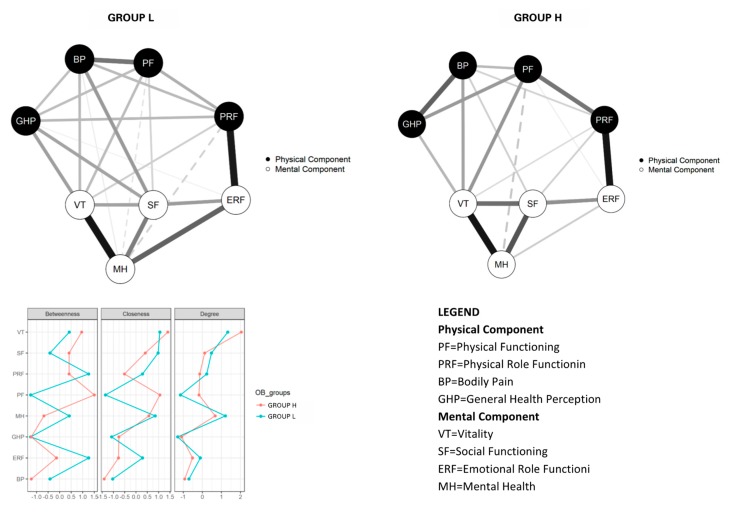
The network of SF-36 quality-of-life domains for patients with obesity walking less than predicted during the six-minute walking test (Group L, on the left), and for patients with obesity walking as far as or farther than predicted during the six-minute walking test (Group H, on the right), and their respective centrality indices (panel C: red line = Group L; blue line = Group H).

**Table 1 nutrients-12-00602-t001:** Demographic and clinical characteristics of patients with obesity walking as far as or farther than predicted during the six-minute walking test (Group H), and of patients with obesity walking less than predicted during the six-minute walking test (Group L). Data are presented as mean ± SD or number (%), as appropriate.

	Group L (*n* = 463)	Group H (*n* = 253)	*t*-Test or Chi-Squared Test	*p*-Value
Gender				
Female	357 (77.1%)	101 (22.1%)	98.1	<0.001
Male	106 (22.9%)	152 (77.9%)		
Age, years	51.1 ± 12.3	50.1 ± 10.5	1.17	0.244
Body weight, kg	112.7 ± 24.8	123.0 ± 24.4	5.36	<0.001
Body mass index, kg/m^2^	41.8 ± 8.0	41.5 ± 6.8	0.46	0.644
Waist circumference, cm	124.2 ±19.3	126.5 ± 18.3	1.51	0.132
Short Form-36				
Physical Functioning	54.9 ± 26.2	71.8 ± 20.8	9.39	<0.001
Physical Role Functioning	51.1 ± 40.7	65.7 ± 38.1	4.64	<0.001
Bodily Pain	50.3 ± 27.4	65.5 ± 25.3	7.31	<0.001
General Health Perception	46.9 ± 19.1	55.7 ± 20.4	5.35	<0.001
Vitality	46.9 ± 20.6	54.1 ± 18.0	4.62	<0.001
Social Functioning	62.4 ± 26.2	67.4 ± 24.4	2.47	0.014
Emotional Role Functioning	60.6 ± 42.3	70.3 ± 39.0	2.97	0.003
Mental Health	60.0 ± 20.6	65.9 ± 17.7	4.04	<0.001

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
