# Peer review of "Quality of Life and Physical Performance in Patients with Obesity: A Network Analysis"

_nutrients, 2020, doi:10.3390/nu12030602_

Round 1
Reviewer 1 Report
- what was the methodology for measuring body weight and height? how many times was the measurement taken? was the patient instructed in what position to be? how the possibility of error during the measurement was excluded
- was the sample size calculated?
- why in Table 1 there are female next to gender while the other parameters have units of measurement, this introduces ambiguity
Author Response
We thank the reviewer for his/her comments. We responded point by point to the comments.
What was the methodology for measuring body weight and height? How many times was the measurement taken? Was the patient instructed in what position to be? How the possibility of error during the measurement was excluded
- We have specified in the text that all data were collected on the second day of admission to the programmes. Specifically, BMI was determined by the standard formula of body weight (kg) divided by height (m2) following measurement of body weight and height using medical weighing scales (Seca Digital Wheelchair Scale Model 664) and a stadiometer (Wall-Mounted Mechanical Height Rod Model 00051A; Wunder), respectively. The scale was calibrated for accuracy by an external accredited laboratory every two months. For the purposes of these measurements, participants were weighted in the morning (12 hours since eating) wearing only lightweight clothes and no shoes and standing with minimal movement with hands by their side. Body weight was measured once for each participant to the nearest 0.1 kg.
Was the sample size calculated?
- We have not calculated the sample size because no clear parameters are furnished in network analysis. According to Epskamp et al. (2018): “Overall, we see that networks with increasing sample size are estimated more accurately. This makes it easier to detect differences between centrality estimates, and also increases the stability of the order of centrality estimates. But how many observations are needed to estimate a reasonably stable network? This important question usually referred to as power-analysis in other fields of statistics (Cohen, 1977) is largely unanswered for psychological networks. When a reasonable prior guess of the network structure is available, a researcher might opt to use the parametric bootstrap, which has also been implemented in bootnet, to investigate the expected accuracy of edge weights and centrality indices under different sample sizes. However, as the field of psychological networks is still young, such guesses are currently hard to come by. As more network research will be done in psychology, more knowledge will become available on graph structure and edge-weights that can be expected in various fields of psychology. As such, power calculations are a topic for future research….”
Why in Table 1 there are female next to gender while the other parameters have units of measurement, this introduces ambiguity
- We have inserted “female” next to “gender” to explain that we have included data about feminine gender. However, we have modified Table 1 to clarify it.
Reviewer 2 Report
This is a very interesting study with real-world implications. The study's design and methodology provided very valid insights into quality of life regarding physical and psychological impacts based on their association with obesity.
The following are areas for revision.
Line(s):
60: Two or more risk factors? What do you mean by "risk factors?" Explain.
60-62: Did your study consider patients with physical disabilities? e.g. using a walker? Where there patients needing assistance/support with walking? Was this an exclusion criteria?
62: What did you classify as "severe psychiatric disorders? besides eating disorders. Given that this study also assess mental health impairments, how did you differentiate between acceptable and non-acceptable psychological/psychiatric impairments?
78-79: Table 1 did not show this - How many patients took breaks/stopped during the assessment?
78-79: Since pulse, respiratory rate, and oxygen levels were checked before and after each test, how did you account for these readings for those patients who rested/stopped during the test? If any patients stopped/rested, did you check pulse, oxygen, and respiratory rates during the resting periods?
136: There is a period (.) between role and in. Should that be there are is that a typo? Please fix. If it should be there then capitalized In.
152: "This could be the attributed to"- Delete "the"
The conclusion could be expanded on in more details.
Author Response
We thank the reviewer for his/her comments. We responded point by point to the comments.
This is a very interesting study with real-world implications. The study's design and methodology provided very valid insights into quality of life regarding physical and psychological impacts based on their association with obesity.
- We thank the reviewer for these positive comments
The following are areas for revision.
Line(s):
60: Two or more risk factors? What do you mean by "risk factors?" Explain.
- We have specified that “risk factors” refer to “cardiovascular risk factors”
60-62: Did your study consider patients with physical disabilities? e.g. using a walker? Where there patients needing assistance/support with walking? Was this an exclusion criteria?
- We have excluded patients using a walker and who needed assistance with walking because they cannot make the 6MWT. We have added this information in the text.
62: What did you classify as "severe psychiatric disorders? besides eating disorders. Given that this study also assesses mental health impairments, how did you differentiate between acceptable and non-acceptable psychological/psychiatric impairments?
- We have specified in the text that we have excluded patients with bulimia nervosa, acute psychotic disorders, and substance use disorders
78-79: Table 1 did not show this - How many patients took breaks/stopped during the assessment?
- As patients using a walker and need assistance/support with walking were excluded, all participants concluded the test without breaks.
78-79: Since pulse, respiratory rate, and oxygen levels were checked before and after each test, how did you account for these readings for those patients who rested/stopped during the test? If any patients stopped/rested, did you check pulse, oxygen, and respiratory rates during the resting periods?
- Unfortunately, we have not collected pulse, oxygen, and respiratory rates. We have added this sentence in the limitation section “…while we have routinely measured pulse, oxygen, and respiratory rates during the 6MWT, we have not collected these data in the data set and therefore we don’t have accurate information about these variables of physical fitness.”
136: There is a period (.) between role and in. Should that be there are is that a typo? Please fix. If it should be there then capitalized In.
- We have corrected the sentence and capitalized “In”.
152: "This could be the attributed to"- Delete "the"
- Done as suggested
The conclusion could be expanded on in more detail.
- We have explained the conclusion in more detail
English language and style are fine/minor spell check require
- The paper has been reviewed by a professional scientific English translator.
Round 2
Reviewer 2 Report
The revisions to the present study have increased the study's validity and added more clarity for the reader. The present study is a very good contribution to the literature.